# DATA AUGMENTATION FOR RUMOR DETECTION USING CONTEXT-SENSITIVE NEURAL LANGUAGE MODEL WITH LARGE-SCALE CREDIBILITY CORPUS

**Sooji Han**\***Jie Gao**\***& Fabio Ciravegna**
Department of Computer Science
The University of Sheffield
211 Portobello, Regent Court, Sheffield, S1 4DP, UK
{sooji.han,j.gao,f.ciravegna}@sheffield.ac.uk

## ABSTRACT

In this paper, we address the challenge of limited labeled data and class imbalance problem for machine learning-based rumor detection on social media. We present an offline data augmentation method based on semantic relatedness for rumor detection. To this end, unlabeled social media data is exploited to augment limited labeled data. A context-aware neural language model and a large credibility-focused Twitter corpus are employed to learn effective representations of rumor tweets for semantic relatedness measurement. A language model fine-tuned with the a large domain-specific corpus shows a dramatic improvement on training data augmentation for rumor detection over pretrained language models. We conduct experiments on six different real-world events based on five publicly available data sets and one augmented data set. Our experiments show that the proposed method allows us to generate a larger training data with reasonable quality via weak supervision. We present preliminary results achieved using a state-of-the-art neural network model with augmented data for rumor detection.

## 1 INTRODUCTION

Research areas that have recently been received much attention in using Machine Learning (ML) and Natural Language Processing for automated rumor and fake news detection (Helmstetter & Paulheim, 2018; Kwon et al., 2017; Shu et al., 2017; 2018; Wong et al., 2018) and fact-checking (Boididou et al., 2018; Vosoughi et al., 2017; Kochkina et al., 2018b). One major bottleneck of state-of-the-art (SoA) ML methods is that they require a vast amount of labeled data to be trained and manual labeling of rumors source on social media requires special skills and time-consuming (Zubiaga et al., 2016). Due to limited labeled training data, existing neural networks (NNs) for rumor detection usually have shallow architecture (Chen et al., 2018; Ma et al., 2016). The scarcity of labeled data is a major challenge of studying rumors on social media (Aker et al., 2017). Another problem is that publicly available data sets for rumor-related tasks such as PHEME data (Kochkina et al., 2018b) suffer from imbalanced class distributions (Liu et al., 2017). Existing methods for handling the class imbalance problem (e.g., oversampling and the use of synthetic data (Xu & Chen, 2015)) may cause over-fitting and poor generalization performance. A methodology for rumor data augmentation with the minimum of human supervision is necessary. Previous studies presented that rumors can evolve into many variants which share similar propagation patterns in their early stage (Maddock et al., 2015; Chen et al., 2018; Baxter et al., 2015; Friggeri et al., 2014). Based on this hypothesis, we argue that enriching existing labeled data with unlabeled source tweets conveying the same or similar meanings is a promising attempt for rumor detection methods that rely on the structure of rumor propagation in social media. In this work, we propose a novel data augmentation method for automatic rumor detection based on semantic relatedness. We exploit a publicly available paraphrase identification corpus as well as context-sensitive embeddings of labeled references and unlabeled candidate source tweets. Pairwise similarity is used to guide the assignment of pseudo-labels to unlabeled tweets. ELMo (Peters et al., 2018), a SoA context-sensitive neural language

---

*indicates equal contribution.

model (NLM), is fine-tuned on a large credibility-focused social media corpus and used to encode tweets. Our results show that data augmentation can contribute to rumor detection with deep learning with increased training data size and a reasonable level of quality. This has potential for further performance improvements using deeper NNs. We present data augmentation results for three events and the performance of a SoA DNN model for rumor detection with augmented data in Section 5.

## 2 DATA

Four publicly available data sets covering a wide range of real-world events on social media as well as a Twitter paraphrase corpus are used for this project.

**SemEval-2015 task 1 data (Xu et al., 2014)** This data is built for paraphrase identification and semantic similarity measurement. This data set is employed in our semantic relatedness method in order to fine-tune optimum relatedness thresholds through pairwise comparisons between the tweet embeddings of labeled reference and unlabeled candidates (see details in Section 4).

**PHEME data set (Kochkina et al., 2018a)** The latest PHEME data (6392078) is used as reference set for data augmentation covering 9 manually labeled rumor events.

**CrisisLexT26 (Olteanu et al., 2015)** This data comprises tweets associated with 26 hazardous events happened between 2012 and 2013. "2013 Boston bombings" data from this data set is used as reference set in this experiment.

**Twitter event data (2012-2016) (Zubiaga, 2018)** This data consists of over 147 million tweets associated with 30 real-world events unfolded between February 2012 and May 2016, among which six events are selected as a pool of candidates source tweets. This covers 'Ferguson unrest', 'Sydney siege', 'Ottawa shooting', 'Charliehebdo attacks', 'Germanwings plane crash', and 'Boston marathon bombings'. We refer to the first five events with reference set generated from PHEME data as **'PHEME5'**. For the 'Boston bombings' event, we generate references from *CrisisLexT26* and a fact-checking website 'Snopes.com'[1] (refer to Section 3).

**CREDBANK (Mitra & Gilbert, 2015)** This large corpus comprises more than 60M tweets grouped into 1049 events, each of which were manually annotated with credibility ratings. This is leveraged to fine-tune ELMo in order to provide better representations for rumor-related tasks (see Section 3).

## 3 METHOD

An overview of our data augmentation method is presented in Figure 1. We exploit a limited amount of labeled data as weak supervision (i.e., references). References are generated separately for PHEME5 and "Boston bombings" data from different data with varying annotations schemes (as mentioned in Section 2). Due to space constraints, we omit the detailed process of reference generation here. Candidate tweets refer to any tweets that report an event of interest. The leftmost box shows how semantic similarity is computed between a given pair of reference and candidate tweets. Firstly, the contextual embedding model (ELMo) is fine-tuned with a domain-specific corpus to learn representations of rumors. Given corpora that contain pairs of tweets, we apply language-based filtering and perform linguistic preprocessing. The preprocessing includes lowercasing, removing 'rt @', URLs[2], and non-alphabetic characters, and tokenization. Tweets with at least 4 tokens are considered to reduce noise (Ifrim et al., 2014). Then, we compute ELMo embeddings of tweets for subsequent semantic relatedness measurement. Cosine similarity between each embeding pair is used as a relatedness measure. We use *SemEval-2015 task 1 data* set as a benchmark for relatedness threshold fine-tuning (see Section 4). Having optimum thresholds, semantic similarity computation is performed for reference-candidate pairs. Rumor and non-rumor source tweets are selected from the candidate pool using the fine-tuned thresholds. In the final step, data collection is performed to retrieve social-temporal context data (typically retweets and replies) for the selected source tweets. Source tweets without contexts are filtered out. We download source tweets for six selected events in the *Twitter event (2012-2016)* and *CREDBANK* using Twitter API[3]. For the *CREDBANK*, we downloaded *77,954,446* tweets (i.e., 97.1% of the original data). After deduplication, the train corpus contains *6,157,180* tweets with *146,340,647* tokens and *2,235,075* vocabularies.

---

[1] https://www.snopes.com/fact-check/boston-marathon-bombing-rumors/

[2] Embedded links may carry critical information for rumor detection, but they are not explored and beyon the scope of this data augmentation task.

[3] An open source tweet collector, available via https://github.com/socialsensor/twitter-dataset-collector

Figure 1: Overview of Data Augmentation.

**Rumor-specific Embedding (ELMo)** Previous research shows that fine-tuning NLMs with in-domain data allows them to learn more meaningful word representations and provides a performance gain (Kim, 2014; Peters et al., 2018). To fine-tune a pretrained ELMo, we generate a data set using the *CREDBANK*. Sentences are shuffled and spit into training and hold-out sets (with a ratio of around 0.02%). We also generate a test set that consists of 6,162 tweets in total using the *PHEME* data. Table 1 shows the number of tweets, tokens and vocabularies in the *CREDBANK* after deduplication. Following the practice in (Perone et al., 2018), a linear combination of the states of each LSTM layer and the token embeddings is adopted to encode tweets. The training corpus is split into small batches with a maximum of 5000 tweets for each batch. Training time took more than 800 hours on a NVIDIA Kepler K40M GPU with less than 10 GiB GPU memory. Since the *CREDBANK* training set is still a relatively small for NLMs, we only fine-tune a pretrained ELMo with 1 epoch to avoid over-fitting. The result shows a large improvement in perplexity on both hold-out and test sets (See Table 2).

Table 1: Statistics of the CREDBANK for fine-tuning ELMo.

|                     | Train       | Hold-out |
|---------------------|-------------|----------|
| # of tweets         | 6,155,948   | 1,232    |
| # of tokens         | 146,313,349 | 27,298   |
| # of vocabularies   | 2,234,861   | 6,517    |

Table 2: Perplexity before and after fine-tuning with one epoch on hold-out set and test set. Reported values are the average of the forward and backward perplexity.

| Data set               | Before tuning | After tuning |
|------------------------|---------------|--------------|
| CREDBANK hold-out set  | 883.06        | 18.24        |
| PHEME test set         | 475.06        | 32.02        |

## 4 EXPERIMENTAL SETUP

**Semantic Relatedness Fine-Tuning** Table 3 compares different models for word representation on the *SemEval-2015 data*. We show the results based on the maximum F-score each model achieves. It shows the effectiveness of our *CREDBANK* fine-tuned ELMo over a pretrained ELMo ("Original (5.5B)") and other SoA word embedding models. To ensure higher quality, we argue that a higher precision is required. Therefore, relatedness thresholds are fine-tuned based on precision achieved by the best-performing model. We highlight some statistics in Tabel 4.

**Data Augmentation** We follow our data augmentation procedure described in Section 3. After performing pairwise similarity computation, a relatedness threshold (*0.8*) is adopted to select rumor source tweets from a pool of candidates. We randomly sample *3*(# of rumor source tweets)* non-rumor source tweets if the score of a candidate tweet is less than 0.3. Sampling more negative examples is an attempt to balance class distributions after source tweets without contexts are removed. This is based on a hypothesis that non-rumors are less likely to have reactions than rumors

Table 3: Comparison of the performance of different models for sentence representation.

| Model                     | F     | P     | R     | Thresh. |
|---------------------------|-------|-------|-------|---------|
| ELMo+CREDBANK (average)   | **0.651** | **0.609** | 0.699 | 0.6526  |
| ELMo+CREDBANK (top)       | 0.627 | 0.566 | 0.703 | 0.6470  |
| ELMo Original 5.5B (average) | 0.628 | 0.587 | 0.675 | 0.6305  |
| ELMo Original 5.5B (top)  | 0.605 | 0.555 | 0.664 | 0.6875  |
| GloVe (twitter.27B.200d)  | 0.508 | 0.342 | **0.989** | 0.5017  |
| Word2Vec (Google News)    | 0.422 | 0.480 | 0.377 | 0.5003  |

Table 4: Results of fine-tuning thresholds based on precision.

| F     | P     | R     | Thresh. | # of rumors (boston) |
|-------|-------|-------|---------|----------------------|
| 0.651 | 0.609 | 0.699 | 0.6526  | 48,086               |
| 0.618 | 0.700 | 0.553 | 0.6911  | 17,205               |
| 0.591 | 0.750 | 0.487 | 0.7083  | 11,212               |
| 0.442 | 0.850 | 0.299 | 0.7602  | 2,617                |
| 0.283 | **0.900** | 0.168 | **0.8018**  | 1,001                |

as they generally draw less attention. In terms of computational performance, ELMo embedding computation and semantic relatedness measurement are performed with CPU. Tweet encoding takes around 10 per second and pairwise comparison takes around 869 pairs per second on average.

## 5   RESULTS AND DISCUSSION

**Data Augmentation**   Before filtering out source tweets without replies, *1,238* rumors and *3,714* non-rumors are collected for "bostonbombings". After filtering, *165* rumors and *228* non-rumors remain. Although the augmented data size is very limited for "bostonbombings", experiments on "sydneysiege" and "ottawashooting" show encouraging results. A total of *25,486* rumors and *76,106* non-rumors are additionally obtained for "sydneysiege", and *21,519* rumors and *62,590* non-rumors are additionally obtained for "ottawashooting". We make our augmented data publicly available [4].

**Rumor Detection**   We conduct rumor detection experiments using two different data sets: (1) *PHEME5*, (2) *PHEME5* with the "bostonbombings" data ("PHEME5+Boston"). We employ (Kochkina et al., 2018b)'s method as a SoA baseline model for rumor detection with slight modifications. For the sake of simplicity, we modify the implementation of "MTL2 Veracity+Detection" for rumor detection only.

Table 5: Rumor detection results for the PHEME5 and augmented data sets.

| Data | F | P | R | Acc. |
|---|---|---|---|---|
| **PHEME5** | 0.535 | 0.580 | 0.497 | 0.707 |
| **PHEME5+ Boston** | 0.493 | 0.580 | 0.429 | 0.696 |

We construct input by using a source tweet and the top (i.e., most recent) 24 replies in this task. We perform leave-one-out cross-validation (LOOCV) on the PHEME5 and augmented data sets. The overall experimental results for rumor detection are presented in Table 5. Table 6 shows LOOCV results. We observe that overall performance decreases with the augmented data (i.e., PHEME5+Boston). The "fergusonunrest" is the most difficult event for a rumor detection model as it has a unique class distribution distinguished from all other events (Kochkina et al., 2018b). It is worth noting that our data augmentation improves the performance of rumor detection on the "fergusonunrest". The completion of data augmentation for events other than "'bostonbombings" has potential to boost overall and per event performance of rumor detection.

Table 6: Cross-validation results. Event column shows an event used as a test set on each iteration.

| Event | Data | F | P | R | Acc |
|---|---|---|---|---|---|
| charliehebdo | PHEME5 | **0.541** | 0.430 | **0.729** | 0.7279 |
| | PHEME5+Boston | 0.507 | **0.452** | 0.576 | **0.753** |
| fergusonunrest | PHEME5 | 0.185 | **0.458** | 0.116 | **0.746** |
| | PHEME5+Boston | **0.318** | 0.417 | **0.257** | 0.726 |
| germanwings | PHEME5 | **0.597** | 0.665 | **0.542** | **0.629** |
| | PHEME5+Boston | 0.530 | **0.682** | 0.433 | 0.610 |
| ottawashooting | PHEME5 | **0.634** | **0.780** | **0.534** | **0.674** |
| | PHEME5+Boston | 0.621 | 0.750 | 0.530 | 0.658 |
| sydneysiege | PHEME5 | **0.550** | **0.717** | **0.446** | **0.688** |
| | PHEME5+Boston | 0.439 | 0.709 | 0.318 | 0.653 |
| bostonbombings | PHEME5+Boston | 0.459 | 0.591 | 0.376 | 0.628 |

## 6   CONCLUSION AND FUTURE WORK

We present a methodology of data augmentation for rumor detection that exploits semantic relatedness between limited labeled data and unlabeled data. This study is part of further research that aims to use a massive amount of publicly available unlabeled Twitter data and the potential of DNNs in a wide range of tasks related to rumors on social media. Our current research has demonstrated the potential efficiency and effectiveness of semantically augmented data in combating the labeled data scarcity and class imbalance problems of publicly available rumor data sets. In future work, we plan to augment data for more events to build comprehensive data sets for rumor detection, and conduct experiments on rumor detection via deep learning. We will evaluate the effectiveness of augmented data in alleviating over-fitting and its usefulness in facilitating deeper NNs for rumor detection. Further experiments will be conducted to examine the generalization of rumor detection models on unseen rumors.

---

[4]https://doi.org/10.5281/zenodo.3249977

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
