# OpenReview forum: "Data Augmentation for Rumor Detection Using Context-Sensitive Neural Language Model With Large-Scale Credibility Corpus"
_ICLR.cc/2019/Workshop/LLD — LLD 2019_

### Official Review · AnonReviewer2 · 2019-04-02
**A useful augmentation method, persuasively tested.**

**Rating:** 4
**Confidence:** 2

**Review:**

This paper addresses the problem of limited data in rumor detection. They augment  data by identifying unlabeled data as paraphrases of  labeled rumors based on semantic similarity. They build a rumor detection model by fine-tuning a pretrained language model.

I recognize space is limited, but a brief explanation of the rumor detection task and  specifics about  the class imbalance would help.

Preprocessing as described removes critical meta information about whether the tweet is citing a particular source (url/rt). I'm skeptical that removing this information is necessary to build a model.

I would like to see some exploration of whether sentence cosine similarity is actually a good metric for semantic similarity. What properties are captured by cosine similarity?

Does this manner of augmenting data create bias towards detecting the same sort of rumors as are in the corpus? That is, will topic be relied on more than other markers of credibility? Perhaps holding out specific events from augmented training data would be a good way to test.

The models that serve as a test bed show a sound methodology, and this paper strikes me as a solid work in progress.

---

### Official Review · AnonReviewer1 · 2019-04-06
**A straight-forward data augmentation method for rumor detection employing ELMo**

**Rating:** 4
**Confidence:** 3

**Review:**

The authors present a data augmentation technique for rumor detection using recently introduced contextualized word representations, like ELMo. Last, they fine-tune them with diverse datasets (tweets at their majority) in order to build rumor-specific embeddings.

The paper is very clear and easy to comprehend. The authors present a very analytical data augmentation technique for the task of rumor detection by employing semantic relatedness fine-tuning on a large Twitter corpus that they collected. This way the effectively address the labeled data scarcity and class imbalance problems.

Pros:
- using state-of-the-art neural language models
- semantic relatedness fine-tuning

Cons:
- not compared with other data augmentation techniques (other than using Kochkina's method)
- considerable time for collecting the data, fine-tuning, and kxn pair comparison

What was the time (as well as resources) requested to fine-tune on the CREDBANK corpus, as well as the time required for the whole process? Will the data and methods be available to the public? Could other methods be used than semantic relatedness? What about involving transfer learning for similar tasks?

---

### Decision · Program_Chairs · 2019-04-08
**Acceptance Decision**

Accept